# Geriatric Neurosurgery in High-Income Developing Countries: A Sultanate of Oman Experience

**Tariq Al-Saadi** [1,2,*], **Abdulrahman Al-Mirza** [3], **Omar Al-Taei** [3] and **Hatem Al-Saadi** [4]

1   Department of Neurology and Neurosurgery, McGill University, Montreal, QC H3A 1Y2, Canada
2   Department of Neurosurgery, Khoula Hospital, Muscat 113, Oman
3   College of Medicine and Health Sciences, Sultan Qaboos University, Muscat 123, Oman
4   Department of General Surgery, Sohar Hospital, Sohar 311, Oman
*   Correspondence: tariq.al-saadi@mail.mcgill.ca

**Abstract:** This study aimed to investigate the prevalence and characteristics of geriatric neurosurgical conditions in the Neurosurgical Department at Khoula Hospital (KH), Muscat, Sultanate of Oman. The majority of various neurosurgical conditions is increasing in elderly patients, which leads to an increase in neurosurgical demand. The aging population has a direct effect on hospital decision-making in neurosurgery. However, limited data are available to assess geriatric neurosurgery in developing countries. A retrospective chart review of geriatric cases admitted to the Neurosurgery Department in KH served as our example of a neurosurgical center in a high-income developing country from January 2016 to 31st December 2019. Patients' demographics, risk factors, diagnosis, Glasgow Coma Scale on arrival, treatment types, and length of stay were recorded. A total of 669 patients who were above the age of 65 years were recruited into our retrospective review. The mean age was 73.34 years in the overall cohort and the male-to-female ratio was (1.6:1). The most common diagnostic category was trauma, which accounted for 35.4% followed by oncology and vascular (16.3% each). Hydrocephalus accounted for 3.7% of the admissions. Most of the patients underwent surgical interventions (73.1%). The associations were significant between the treatment types (surgical vs. conservative), Length of Stay, and the GCS on arrival ($p < 0.05$). In conclusion, the trend of geriatric neurosurgery is increasing in developing countries. The most common reason for admission to the neurosurgical ward was Traumatic Brain Injury. Special care must be taken when dealing with geriatric neurosurgical cases and a more holistic approach is needed.

**Keywords:** geriatric; neurosurgery; elderly; surgical intervention; Oman

## 1. Introduction

As with developed countries, improvements in living conditions and advancements in healthcare have yielded an increasing life expectancy and the number of aging populations [1]. This increase has been associated with an increase in the prevalence of non-communicable diseases including neurological disorders. Such an increase comes with an increase in needed resources, which presents economic challenges in countries with limited resources. Neurosurgery as a specialty is a medical field that specializes in the prevention, diagnosis, and surgical management of brain disorders as well as rehabilitation. People aged 65 years and older make up 11–14% of the general population. This increase in the elderly population is a feature of industrialized countries, for instance, in Sweden the elderly account for 16.4% of the entire population. This is in contrast to India, where the elderly makes up 3.1% of the population [2]. With the rise in the elderly population, there is an increasing demand in the healthcare sector, including neurosurgical referrals [2]. This was seen in a study carried out by Gonzales Bonet in Spain showing an increase in geriatric neurosurgical admission by 77.5% [3]. Elderly patients are not exempt from needing neurosurgical procedures, but rather they require extra measures prior to the

surgery. The prevalence of various neurosurgical diseases, for instance, primary malignant tumors, subdural hematomas, and degenerative spine diseases, is increasing in elderly patients, which leads to an increase in neurosurgical demand in the elderly [4]. A greater percentage of neurosurgeons will involve these patients [2].

Almost 10 million people are affected annually by traumatic brain injury (TBI), including the elderly. In Japan, about 24% of the population is elderly and this is projected to rise to 30% by 2030 [5]. Correspondingly, In the United States, the elderly population accounts for 15% of the population, and this is anticipated to reach 23% by 2030 [1]. Over the past several decades, the evolution of medical technologies, equipment, and neurosurgical procedures and techniques have enlarged and refined the neurosurgical practice. This evolution resulted in the enlargement of conditions that could be managed by neurosurgical techniques [6]. Not only that but, this evolution reduced the mortality and morbidity of the comorbid elderly patient and provided a safer and more efficient approach [3]. One of these difficulties that arise from dealing with elderly patients is a lack of communication and confusion. Additionally, due to morbidities, the elderly present with other medical conditions make it harder in to manage their neurological condition. In addition, the risk of operative intervention and anesthesia is also higher in the elderly [2]. However, local evidence is needed to assess the current burden of this age group on neurosurgical care as well as other interventions that can improve their survival.

In Oman, a high-income developing country, there was an increase in the elderly population from 2013 to 2017 by 15.4%. The aim of this study was to retrospectively analyze the prevalence, outcome, and follow-up of neurosurgical procedures in elderly patients in a tertiary hospital in Oman as an example of a high-income developing country experience. Oman is considered to have one of the best well-rounded healthcare systems according to the World Health Organization (WHO)'s reports [7]. A study investigating the public satisfaction with Oman's healthcare system among the Omani population revealed that 64.1% were satisfied with the quality of the general health services that the government offers [7]. Khoula Hospital (KH) is one of the three tertiary hospitals in Oman, and it is considered a leading center in neurosurgical services at the national level, with around 1400 patients admitted per year, 10% of which are diagnosed with brain and spinal tumors [8]. In this study, we chose a cut-off of 65 years, [9] considering the increase in life span throughout the last decades as well as the improvement in the quality of life of older people who nowadays remain socially active until the age of ≥70 years.

## 2. Methods and Study Design

### 2.1. Study Group

This is a retrospective study conducted at KH in Muscat, Sultanate of Oman as an example of a high-income developing country. The study was approved by the Research Ethical Committee at KH and the Ministry of Health in the Sultanate of Oman (PR0122020072). Medical records of 669 patients admitted to the neurosurgical department only who were above the age of 65 and from the period of January 2016 to 31 December 2019 were included. The study included both Omani and non-Omani patients. Patients with the following features were excluded: patients less than 65 years old, those with non-neurosurgical conditions, and patients admitted outside the study period (from 1st January 2016 to 31 December 2019). Spinal cases admitted to Orthopedic Department were also excluded (as spinal cases are divided between the Neurosurgery and the Orthopedic Department based on the on-call team). Thirty-one patients were excluded in the study period due to a lack of diagnostic information, missing fields on the length of stay, medication, and some fields that were duplicated.

### 2.2. Data Collection

Data that were obtained from the health information system included patient demographics, presenting symptoms, pre-operative medical conditions, post-operative complications, previous surgical history, preoperative and postoperative Glasgow Coma Scale

(GCS), radiological findings, indication for surgery, diagnosis, clinical outcome, length of hospital stay (LOS), length of ICU admission, and the treatment proposed. Data on treatment modalities were collected. Then the information was classified into continuous and categorized variables and analyzed accordingly.

### 2.3. Data Analysis

The research database was analyzed and processed using the statistical package for the social sciences (SPSS) software (version23). The categorized variables were cross-tabulated using frequency tables and descriptive linear charts. The Chi-square test was used to obtain the significance of the association between categorized variables, using a $p$-value $\leq 0.05$ as the cut-off for significance. The numerical variables were summarized by their medians, means, and ranges, and the categorical variables were described by their counts and relative frequencies. All the $p$ values were 2-sided, and a $p$-value $< 0.05$ was significant in all the analyses.

### 3. Results

The demographic characteristics of the included cases in the present study are shown in Table 1. We had a total of 669 patients admitted to the neurosurgical department at KH in Muscat, the capital city of the Sultanate of Oman, in a four-year period (from 2016 to 2019). The year 2019 accounted for the highest number of admitted patients (30%). The mean age of the included patients was 73.34 years (65–98 years). The male-to-female ratio was (1.63:1). Most of the patients had GCS scores of 14–15 (72.3%). The most common diagnostic category was trauma, accounting for 35.4% of the study cohort followed by oncology and vascular (16.3% each). Most of the patients underwent surgical intervention (73.1%), and 77% of the patients stayed in the hospital for more than 15 days.

**Table 1.** Demographic characteristics of the geriatric patients.

| Category | Number of Patients (%) |
| --- | --- |
| **Number of patients admitted each year** | |
| 2019 | 202 (30.0%) |
| 2018 | 172 (25.7%) |
| 2017 | 154 (23%) |
| 2016 | 141 (21.3%) |
| **Total** | **669** |
| **Age** | |
| ≥75 | 414 (61.9%) |
| <75 | 255 (38.1%) |
| **Gender** | |
| Female | 255 (38.1%) |
| Male | 414 (61.9%) |
| **GCS on arrival** | |
| 15–14 | 484 (72.3%) |
| 13–12 | 49 (7.3%) |
| 9–11 | 36 (5.3%) |
| <8 | 100 (15%) |
| **Diagnostic category** | |
| Oncology | 109 (16.3%) |

**Table 1.** *Cont.*

| Category | Number of Patients (%) |
|---|---|
| **Number of patients admitted each year** | |
| Trauma | 237 (35.4%) |
| Vascular | 109 (16.3%) |
| Spine and Peripheral nerve diseases | 176 (26.3%) |
| Infection and functional | 13 (2%) |
| Hydrocephalus | 25 (3.7%) |
| **Type of treatment** | |
| Surgical | 489 (73.1%) |
| Conservative | 180 (26.9%) |
| **Length of stay (LOS)** | |
| $\leq$15 days | 515 (7%) |
| >15 days | 154 (23%) |

Figure 1 represents the trend of the total number of admitted patients each year with the corresponding diagnostic category. As seen in the graph, trauma cases were continuously raising during the study years with the highest number of cases in 2019. Spinal and peripheral nerve cases showed a steady growth between the study years, as well as the hydrocephalus, infection, and functional cases, but to a lesser extent. Oncology and vascular cases showed a fluctuating pattern among the study years.

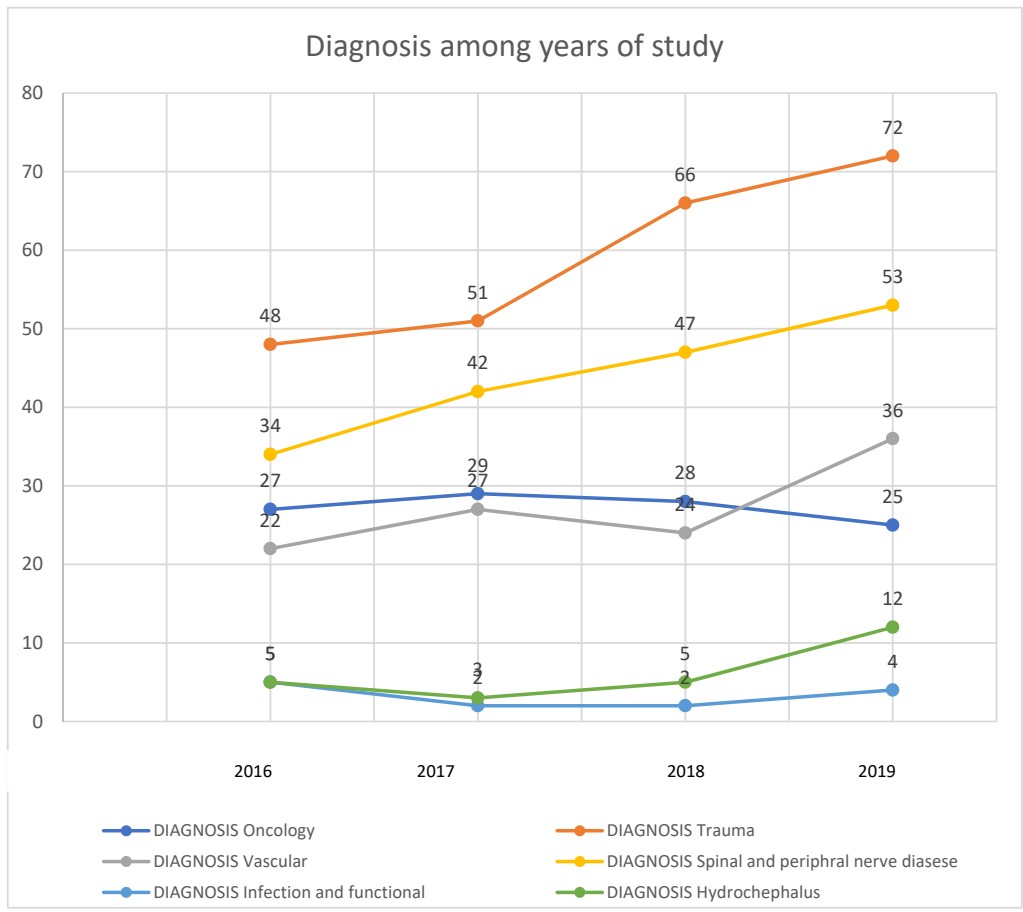

**Figure 1.** Total number of admitted patients in each year with the corresponding diagnostic category.

The association between the age group and the other variables (diagnosis type, LOS, and treatment type) is demonstrated in Table 2. There was a significant difference between the age of patients above and below 75 years and the diagnosis type (trauma vs. non-trauma), in which the traumatic type of injury was found to affect the older age group (more than 75 years) ($p < 0.005$). Additionally, it shows that there was no association between age above and below 75 years and the LOS (15 days as a cut-off value), where both age groups had a similar period of stay at the hospital ($p = 0.257$). There was no significant association between age above and below 75 years and type of intervention, whether surgical or conservative management ($p = 0.329$).

**Table 2.** The association between the age group (more and less than 75 years) and other variables.

| Variable | *p*-Value |
|:---:|:---:|
| **Diagnosis type** | |
| Trauma | <0.005 |
| Non Traumatic | |
| **Length of stay (LOS)** | |
| ≤15 days | 0.257 |
| >15 days | |
| **Treatment types** | |
| Surgical | 0.329 |
| Conservative/observation | |

Table 3 illustrates the relationship between the treatment types (surgical vs. conservative) and the other variables. As shown, there was a significant relationship between the type of treatment and the LOS, in which the group who underwent surgical intervention was found to have more LOS period compared with the conservative group ($p < 0.005$). It was also shown that there was no association between type of and the diagnosis types ($p = 0.217$).

**Table 3.** The association between type of treatment (surgical vs. conservative) and other variables.

| Variable | *p*-Value |
|:---:|:---:|
| **Diagnosis type** <br> Trauma <br> Non Traumatic | 0.217 |
| **Length of stay (LOS)** <br> ≤15 days <br> >15 days | 0.005 |

The relationship between the GCS on arrival (less and more than 8) and the other variables is represented in Table 4. There was an association between the GCS and the type of treatment that the patient received (surgical vs. conservative). The association between the GCS and the LOS was also significant, in which the group who stayed for a shorter period at the hospital were found to have higher GCS on arrival compared with another group ($p < 0.005$). There was no association between the GCS (less and more than 8) and the diagnosis type ($p = 0.896$).

**Table 4.** The association between the GCS on arrival and other variables.

| Variable | *p*-Value |
|---|---|
| **Diagnosis type** | |
| Trauma | <0.005 |
| Non Traumatic | |
| **Length of stay (LOS)** | |
| ≤15 days | <0.005 |
| >15 days | |
| **Treatment types** | |
| Surgical | 0.896 |
| Conservative/observation | |

## 4. Discussion

It is estimated that the elderly population will account for 18% of the world population by 2060 [10] indicating an increase in life expectancy due to improvements in living conditions [3]. This illustrates a crucial health challenge due to the burden on the health service and on the financial concerns involved in the treatment of the elderly [11,12]. In our cohort, the average age of patients above 75 was almost two-thirds of the study sample. This is because the Omani population's life expectancy for both sexes is almost 77.95 years [13]. The Omani national census found that the geriatric population in Oman has increased, which is similar to the findings in a study carried out by S. Chibbaro [5]. The same census found that the female ratio in Oman was 180.8 males per 100 females [6]. In this current study, male patients accounted for 61.9% of the study sample in comparison with other studies in the literature with many female patients.

Middle-aged males are preponderantly affected due to their high contribution to outdoor activities and frequent engagement in motor vehicle accidents and sports injuries [14]. The most common diagnosis in our cohort was TBI (35.4%), followed by spine and peripheral diseases (26.3%), and then oncological and vascular conditions (16.3% each). These findings are similar to findings from other studies where TBI and spinal operations for degenerative changes were the most common indications for admission to neurosurgery units in the geriatric population [5,7]. TBI presents more commonly in the elderly population as minor and major domestic falls account for the majority of injuries in this population. An analysis carried out by Yokobiri et al. found that non-traffic-related accidents, including falls, have tremendously increased over the past year along with a significant reduction in road traffic accidents (RTAs) [14].

The findings of this study showed that patients over the age of 75 were found to be affected with TBI more frequently compared with patients less than 75 years old. This was supported by a study done by Hilaire J. Thompson at el. in which adults aged 75 and older had the highest rates of TBI-related hospitalization and death [15]. Traumatic injuries in elderly patients have different mechanisms and outcomes when compared with non-elderly patients. Numerous factors increase the risk of TBI in elderly patients. For instance, atrophy of the brain, which is associated with increasing age, results in extending the distance between the brain and the skull, resulting in a dura vessel more prone to shearing damage. Additionally, dura becomes more adherent to the skull with increasing age. Besides that, the use of aspirin and anticoagulants for chronic disease management puts elderly patients at risk of TBI and minor injury [16]. As seen in our cohort, TBI is continuously rising each year with increasing age. Similar results were also found in a study by George Stranjails et al. in which TBI in Greece showed an upward trend across the period 2010–2018 [17]. Elderly patients with TBI have shown that they have less chance of surviving neurosurgery for extradural and subdural hematoma compared with those aged under 65 [18]. Adding to that, Oman has a high rate of road traffic accidents and the basics of managing epidural hematomas vary case by case, which include primary survey with simultaneous initiation of neuroprotective measures, empiric ICP management, urgent craniotomy, hematoma/clot

evacuation, and prevention of complications in brain injuries (e.g., anticoagulant reversal to prevent hematoma expansion). Conservative management with close observation and serial CT scan be considered for a small, asymptomatic EDH [1,4].

Patients with TBI over the age of 70 years and with a GCS of 14 are found to have increased mortality, in contrast with patients aged less than 70 with lower GCS. This is because the elderly usually present with higher GCS compared with younger with corresponding injuries [19]. The current study demonstrated the fact that the lower the GCS, the longer the length of hospital stay will be. It also signifies that patients who underwent surgical interventions had longer LOS compared with those managed conservatively. This may be due to postoperative complications that might extend the LOS through management of the complications (e.g., reoperations, chest infections, urinary infections, and other medical-related conditions); therefore, providers should focus on preventing and managing complications to improve overall efficiency [19].

The elderly population has affected the epidemiology of traumatic Spinal Cord Injuries (SCI), causing a surge in fall-related cervical injuries over the past three decades. However, there is a debate in the literature regarding the impact of age on recovery at the time of impact. Moreover, there is a scarcity of information regarding the economic effect of the increased percentage of elderly patients surviving traumatic SCI [15].

According to a study carried out by Romain Pirracchio, the prevalence of intracranial tumors increases proportionally with age at a maximum of 75 years of age. This was particular for gliomas and meningiomas [20]. A previous study conducted at KH showed that the prevalence of brain and spinal tumors accounted for 9.6% of admitted cases to the Department of Neurosurgery, of which brain tumors accounted for 91% of the cases [20]. The same study reported that the two most common tumor pathologies were meningioma and metastatic tumors [9]. A recent study published regarding geriatric neuro-oncology in Oman found that meningiomas were the most common tumor at KH by 52.8% [21] while the commonest vascular neurosurgical admission to KH was Intracerebral hemorrhage (ICH). Besides, those degenerative spinal conditions were the most frequent cause of admission in KH with almost 91% of all spinal causes. In the current study, 16.3% of geriatric patients were admitted to the department of Neurosurgery in KH for oncological conditions. On the other hand, studies regarding neurological tumors illustrate that patients aged more than 75 years and diagnosed with glioblastoma have a lower survival rate compared with patients aged less than 75 [22].

## 5. Limitations

There were several limitations in this study. It was a retrospective, single-centered, cross-sectional study carried out over four years. Therefore, several confounding factors exist, such as the availability of diagnostic imaging facilities, advancements in modern medical technology, and improvements in the intensive care unit. Further studies in the future are recommended, keeping in mind the considerations of the current limitations. Follow-ups were not involved in this study. Adding to that, the implementation of telemedicine for following up with patients is part of the future plan given the advantages that telemedicine has to offer.

## 6. Conclusions

The trend of geriatric neurosurgery is increasing in developing countries. The most common reason for admission to the neurosurgical ward was TBI. The length of stay was longer in the surgical intervention group compared with the conservative group. Special care must be taken when dealing with geriatric neurosurgical cases and with a more holistic approach.

**Author Contributions:** Conceptualization, T.A.-S.; methodology, T.A.-S.; software, A.A.-M.; validation, T.A.-S., A.A.-M. and O.A.-T.; formal analysis, A.A.-M. and O.A.-T.; investigation, A.A.-M. and O.A.-T.; resources, A.A.-M. and O.A.-T.; data curation, A.A.-M. and O.A.-T.; writing—original draft preparation, A.A.-M. and O.A.-T.; writing—review and editing, H.A.-S., T.A.-S., A.A.-M. and O.A.-T.; visualization T.A.-S., H.A.-S., A.A.-M. and O.A.-T.; supervision, T.A.-S.; project administration, T.A.-S. All authors have read and agreed to the published version of the manuscript.

**Funding:** This research received no external funding.

**Institutional Review Board Statement:** The study was conducted according to the guidelines of the Declaration of Helsinki, and approved by the Ethics Committee of Khoula Hospital/Ministry of health (PRO122020072).

**Informed Consent Statement:** Not applicable.

**Data Availability Statement:** From medical records of patients from the "Al-Shifa Health Information System" of Ministry of Health in Sultanate of Oman used in Khoula Hospital.

**Conflicts of Interest:** The authors declare no conflict of interest.

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
