# Peer review of "Geriatric Neurosurgery in High-Income Developing Countries: A Sultanate of Oman Experience"

_2673-5318, doi:10.3390/psychiatryint3040021_

Round 1

Reviewer 1 Report

1-      The author should use abbreviations for the first sentence in the article, so the abbreviation Khola Hospital(kH) should be placed in line 13 and deleted in line 18.

2-    Please write the full sentence first and then its abbreviation However, in this article, abbreviations GSR, TBI, and GCS are used, and these abbreviations are ambiguous for the reader, and the full sentence of the abbreviation should be written before these words, and abbreviations should be used in the rest of the article. For example, lines 19, 26 and 28.

3-    Of should be delet in first line 109

Author Response

Reply to reviewer’s comments:

Author's Response: We appreciate the editor's favorable commentary on our paper and agree with his/her synopsis. We also thank the reviewers for taking care of and reviewing our manuscript and providing constructive feedback. We took into consideration all comments and rewrote the manuscript as suggested.

We are submitting the revised version of the manuscript looking for publication in your esteemed journal.

English Language and Style

(x) English language and style are fine/minor spell check required

The authors reply: “ All spelling and grammar have been checked and revised.”

Reviewer 1:

  • The author should use abbreviations for the first sentence in the article, so the abbreviation Khola Hospital(kH) should be placed in line 13 and deleted in line 18.

The authors reply: We thank the reviewer for rarely this valid point. The above comment has been edited accordingly and highlighted.

  • Please write the full sentence first and then its abbreviation However, in this article, abbreviations GSR, TBI, and GCS are used, and these abbreviations are ambiguous for the reader, and the full sentence of the abbreviation should be written before these words, and abbreviations should be used in the rest of the article. For example, lines 19, 26 and 28.

The author's reply: The reviewer highlighted a good comment. We have edited the following abbreviation as per your request.

  • Of should be delete in first line 109. The author's reply: This is a relevant comment raised by the reviewer. We have deleted “Of” in line 109.

Reviewer 2 Report

Very thoughtful article. But this is a general overview of the population and the relevance Neurosurgery plays in developing countries. There is no conclusion to this brilliant article. Why & How ? Was this an oversight? Please include a 2 paragraph conclusion. Summarize all relevant information in your conclusion

Author Response

Reply to reviewer’s comments:

Author's Response: We appreciate the editor's favorable commentary on our paper and agree with his/her synopsis. We also thank the reviewers for taking care of and reviewing our manuscript and providing constructive feedback. We took into consideration all comments and rewrote the manuscript as suggested.

We are submitting the revised version of the manuscript looking for publication in your esteemed journal.

Reviewer 2:

Very thoughtful article. But this is a general overview of the population and the relevance Neurosurgery plays in developing countries. There is no conclusion to this brilliant article. Why & How ? Was this an oversight? Please include a 2 paragraph conclusion. Summarize all relevant information in your conclusion.

The author's reply: Dear commenter thank you for your insight on our paper we have added a paragraph of conclusion summarizing the main findings of this article and a limitation section on what needs to be done for future work. Line: 250-264
